# Generalization of contextual fear is sex-specifically affected by high salt intake

Jasmin N. Beaver[1,2], Brady L. Weber[1,2], Matthew T. Ford[1], Anna E. Anello[1,2], Kaden M. Ruffin[1], Sarah K. Kassis[1,2], T. Lee Gilman[1,2,3]*

1 Department of Psychological Sciences, Kent State University, Kent, Ohio, United States of America, 2 Brain Health Research Institute, Kent State University, Kent, Ohio, United States of America, 3 Healthy Communities Research Institute, Kent State University, Kent, Ohio, United States of America

* lgilman1@kent.edu

**Data Availability Statement:** The data are accessible at the following DOI: 10.17605/OSF.IO/E5K2M.

## Abstract

A hallmark symptom of many anxiety disorders, and multiple neuropsychiatric disorders more broadly, is generalization of fearful responses to non-fearful stimuli. Anxiety disorders are often comorbid with cardiovascular diseases. One established, and modifiable, risk factor for cardiovascular diseases is salt intake. Yet, investigations into how excess salt consumption affects anxiety-relevant behaviors remains little explored. Moreover, no studies have yet assessed how high salt intake influences generalization of fear. Here, we used adult C57BL/6J mice of both sexes to evaluate the influence of two or six weeks of high salt consumption (4.0% NaCl), compared to controls (0.4% NaCl), on contextual fear acquisition, expression, and generalization. Further, we measured osmotic and physiological stress by quantifying serum osmolality and corticosterone levels, respectively. Consuming excess salt did not influence contextual fear acquisition nor discrimination between the context used for training and a novel, neutral context when training occurred 48 prior to testing. However, when a four week delay between training and testing was employed to induce natural fear generalization processes, we found that high salt intake selectively increases contextual fear generalization in females, but the same diet reduces contextual fear generalization in males. These sex-specific effects were independent of any changes in serum osmolality nor corticosterone levels, suggesting the behavioral shifts are a consequence of more subtle, neurophysiologic changes. This is the first evidence of salt consumption influencing contextual fear generalization, and adds information about sex-specific effects of salt that are largely missing from current literature.

## Introduction

Excessive dietary salt intake occurs across nations and cultures [1, 2]. While salt's practical uses include food preservation and flavor enhancement, consumption of too much salt has been associated with increased risk for cardiovascular diseases [3–6]. Indeed, average estimates of global salt intake indicate people consume 3,950 mg of sodium (the component of NaCl that has been associated with multiple negative health outcomes) [3, 7, 8]. When put in the context that, according to the World Health Organization, adults should consume less than 2,000 mg

**Funding:** Funding for this work was provided by Kent State University, and the Applied Psychology Center in the Department of Psychological Sciences at Kent State University.

**Competing interests:** The authors declare that no competing interests exist.

of sodium, this overconsumption of salt becomes quantifiably apparent [9]. It is estimated that reducing consumption of salt can attenuate risk for cardiovascular disease by ~20% [5, 10], meaning dietary salt consumption has a profound physiological influence.

Globally, cardiovascular diseases and neuropsychiatric disorders, particularly disorders involving anxiety symptoms, are often comorbid [11–15] (see also reviews [16, 17]). Increasing literature suggests dietary components such as excess sugar and/or fat are associated with both metabolic conditions like insulin resistance and type II diabetes, as well as with neuropsychiatric conditions (e.g., depression) and diseases (e.g., Alzheimer's) [18–22]. Trait anxiety appears largely unaffected by excess salt consumption [23–27] (reviewed in [28]). Far less understood is how excess salt intake might affect neuropsychiatric symptoms like generalized (i.e., non-specific, state) anxiety, though some researchers have begun to recognize this possibility [29, 30]. We hypothesized that excessive consumption of dietary salt would enhance fear generalization in rodents, a possibility that has not yet been explored.

Fear generalization is an established neurobehavioral phenomenon that occurs across species, and studies are making considerable progress in identifying brain regions critical to this process [31–34]. These include specific subregions of the cingulate and prefrontal cortices, hippocampus, amygdala, and brainstem nuclei (e.g., locus coeruleus). Growing evidence indicates the hippocampus and amygdala are affected by excess salt consumption in rodents [24, 26, 35–41], including: reductions in nicotinic receptor levels [35]; increased markers of oxidative stress [24, 26, 37, 41], neuronal activation [42], and neuroinflammation [42]; reduced long-term potentiation [40]; and increased microvasculature leakiness [36]. Other brain regions remain underexplored. This evidence, combined with human evidence of comorbidity between anxiety disorders and cardiovascular diseases, supports our hypothesis–at least based upon male data–that fear generalization is augmented by high salt intake. Unfortunately, both the scant literature on behavioral effects of increased salt intake, and the larger body of work on the neurocircuitry of fear generalization, have focused almost exclusively on male rodents. Here, we included both sexes in our studies, but given the current state of the literature we could not formulate data-based *a priori* hypotheses specifically regarding how female fear generalization behavior might differ from male fear generalization under conditions of excess salt intake.

## Materials and methods

### Animals

Mice of both sexes on a C57BL/6J background were purchased from Jackson Laboratory (Bar Harbor, ME) and bred in-house. All mice were at least 9 weeks old prior to commencing experiments. At the start of diet manipulation, mice were singly housed to facilitate accurate quantification of diet and water consumption relative to the mouse's body weight. Prior to experiments, all mouse cages contained Nestlets (Ancare, Bellmore, NY) and huts; at the time of singly housing for diet manipulation, two additional pieces of enrichment (e.g., swing, wood block, etc.) were added to each cage. Cages contained 7090 Teklad Sani-chip bedding (Envigo, East Millstone, NJ) and were housed in a 12:12 light/dark cycle room, with lights on at 07:00, and temperature maintained at 22 ± 2˚C.

Prior to diet manipulation, mice were fed LabDiet 5001 rodent laboratory chow (LabDiet, Brentwood, MO) *ad libitum*. At the time of diet manipulation, mice were pseudorandomly assigned to either a control diet (0.4% NaCl w/w; D17012, Research Diets, Inc., New Brunswick, NJ) or a high salt diet (4.0% NaCl; D17013; Fig 1A). These diets were available *ad libitum*, and water was always available *ad libitum*. Twice weekly, measurements of diet and water consumption plus body weight were taken. Refer to Supporting Information for statistics and graphs on average water consumed per day (S15 and S19 Tables; S9 and S13 Figs); average

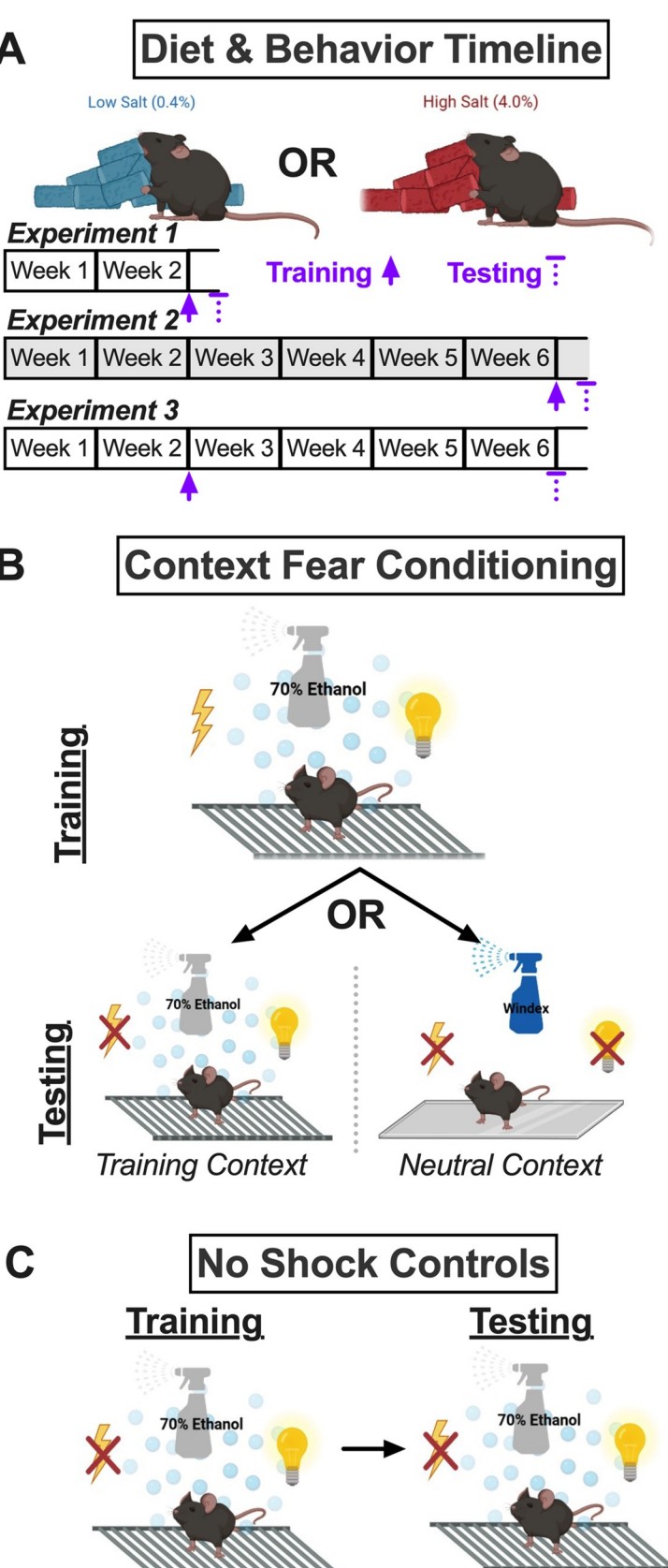

**Fig 1. Study timeline and fear conditioning procedures.** A) Mice were assigned to either a control (0.4% NaCl) or high salt (4.0% NaCl) diet for two or six weeks. Experiment 1 involved diet manipulation for two weeks, at which time mice underwent context fear training, followed 48 h later by context fear testing. Experiment 2 (grey shading) involved diet manipulation for six weeks, at which time mice underwent context fear training, followed 48 h later by context fear testing. Experiment 3 involved diet manipulation for two weeks, at which time mice underwent context fear training, followed four weeks later (while continuing the same diet manipulation) by context fear testing. B) Mice trained with a mild foot shock in the Training Context were tested 48 h (Experiments 1, 2) or four weeks (Experiment 3) later in either the Training Context or the Neutral Context. The Training Context included a metal grid floor, visible illumination, patterned background, and 70% ethanol scent. The Neutral Context included a smooth acrylic floor, infrared illumination, no background, and Windex scent. C) Control no shock mice were exposed to the Training Context, but were never administered a foot shock. These mice were tested at the same timelines for Experiments 1–3 in the Training Context, to assess for any baseline influences of diet consumption on fear behavior.

food consumed per day (S16 and S20 Tables; S10 and S13 Figs); weekly body weight changes (S17 and S21 Tables; S11 and S14 Figs); raw body weights (S18 and S22 Tables; S12 and S14 Figs); average NaCl consumed per day (S23 and S28 Tables; S15 and S20 Figs); average NaCl consumed as percentage of body weight (S24 and S29 Tables; S16 and S20 Figs); weekly ratio between water:NaCl consumption (S25 and S30 Tables; S17 and S21 Figs); average kcal consumed per day (S26 and S31 Tables; S18 and S22 Figs); and average kcal consumed as percentage of body weight (S27 and S32 Tables; S19 and S22 Figs).

Diet manipulations lasted for two weeks (Experiment 1; Fig 1A) or six weeks (Experiments 2 and 3; Fig 1A). All procedures were approved by the Kent State University Institutional Animal Care and Use Committee (protocol 536 LG 22–14), and adhered to the National Research Council's Guide for the Care and Use of Laboratory Animals, 8th Ed. [43]. Every measure was taken to minimize suffering.

## Experimental timelines

Experimental timelines are graphically illustrated in Fig 1A. For Experiment 1, context fear training occurred after two weeks of diet manipulation, and context fear testing occurred 48 h after training. For Experiment 2, training occurred after six weeks of diet manipulation, and testing occurred 48 h after training. For Experiment 3, training occurred after two weeks of diet manipulation, and testing occurred four weeks after training (a standard time frame for natural development of fear generalization in mice [44]).

**Context fear conditioning.** Females and males always underwent behavior separately. Fear conditioning took place in identical Coulbourn Instruments chambers (7 in D × 7 in W × 12 in H; Allentown, PA) composed of two opposing aluminum walls and two Plexiglas walls surrounding a stainless-steel shock grid floor. Each chamber was located within its own sound-attenuating enclosure, and had a camera mounted to the top for real-time measurements of freezing behavior (absence of all movement except that necessary for breathing) using FreezeFrame (v. 5.201, Coulburn Instruments).

**Context fear training.** The Training Context (Fig 1B) consisted of the shock grid floor, a patterned background, visible illumination, and scent cue (70% ethanol). Training occurred at either the two week or six week time point (Fig 1A). Training involved exposure to five, 1 s, 0.8 mA scrambled foot shocks pseudorandomly administered during a 6 min training session at 137, 186, 229, 285, and 324 s. Training with five foot shocks is optimal for eliciting time-dependent contextual fear generalization in rodents [44–46]. Freezing behavior was quantified as the average percent freezing across the first six 5 s bins (i.e., 30 s) that occurred immediately after the 5 s bin containing the foot shock.

**Context fear testing.** Testing occurred at either the two (Experiment 1) or six week (Experiments 2 and 3) time point (Fig 1A). Mice were tested either in the Training Context, or

in a Neutral Context which consisted of a smooth floor, no background, only infrared illumination, and a different scent cue (Windex®, SC Johnson, Racine, WI; Fig 1B). All mice were only ever tested in a single context, and testing lasted for 10 minutes, with the average percentage freezing occurring during minutes two through six being quantified and analyzed [46]. Foot shocks were never administered during testing.

**No shock controls.** Control 'no shock' mice were run concurrently in every cohort of animals (Fig 1C) for every Experiment. These no shock mice were exposed to the Training Context for 'training', but no foot shock was ever administered. Likewise, control no shock mice were tested in the Training Context, to evaluate how diet manipulation and the Experiment timelines influenced freezing behavior in the absence of any contextual fear conditioning.

## Serum measurements

Forty-eight hours after testing for context fear expression, mice were briefly anesthetized with isoflurane then rapidly decapitated for collection of trunk blood. Trunk blood was allowed to clot at room temperature for 30 min, then was spun at 3500 rpm for 1 h at 4˚C. Serum was then removed from tubes and aliquoted into two vials–one for measuring serum osmolality, the other for measuring serum corticosterone. Serum osmolality was measured immediately using an Osmo1 single-sample micro-osmometer (Advanced Instruments, Norwood, MA). Prior to every serum batch, a Clinitrol$^{TM}$ 290 reference solution (Advanced Instruments) from a new ampule was measured to confirm accurate readings. Samples for serum corticosterone were stored immediately at −80˚C until analysis with a corticosterone ELISA kit (Enzo Life Sciences, Inc., Farmingdale, NY) using their small volume protocol. Serum corticosterone levels were log-transformed to account for positive skewness [27, 47, 48]. Log-transformed serum corticosterone levels are, for brevity, hereafter referred to as 'cort' levels.

## Statistical analyses

Data were analyzed with GraphPad Prism 9.4.0 (GraphPad Software, San Diego, CA) and IBM SPSS Statistics 28.0.0.0 (IBM, Armonk, NY). Our *a priori* significance threshold was set at p<0.05. Analyses were made within each Experiment. Repeated measures data (diet/water consumption, salt or kcal calculations, body weight, context fear training) were analyzed using 3-way repeated measures ANOVAs within individual sexes ([repeated measure, e.g., body weight] × diet × context) and pairwise comparisons with Bonferroni correction. Greenhouse-Geisser corrections were employed for within-subjects analyses. Measurements of contextual fear expression, serum osmolality, and serum corticosterone across contexts were analyzed with a 3-way ANOVA (diet × sex × context) and Holm-Šídák's post-hocs. Control no shock fear behavior data were analyzed separately, given these were purposely included as negative controls and not intended to be analyzed in comparison to experimental groups. These no shock control data were analyzed on their own with 3-way ANOVAs (diet × sex × context) and Holm-Šídák's post-hocs, or with repeated measures 3-way ANOVAs within each sex ([repeated measure, e.g., body weight] × diet × sex) plus pairwise comparisons with Bonferroni correction, and also using Greenhouse-Geisser corrections for within-subjects analyses. All ANOVA statistics are reported in S1–S32 Tables. The criterion to exclude outliers was *a priori* assigned as being >5 standard deviations ± mean. Numbers of mice per sex/shock/diet/context condition varied, and exact numbers are present in the respective figure legends for each group of data. Broadly: no shock fear behavior n = 7–9; shock fear behavior n = 7–10; shock serum osmolality n = 8–9; shock serum corticosterone n = 7–9. Details of outliers that were excluded (training baseline, n = 1; no shock testing, n = 1; shock testing, n = 4; osmolality, n = 1; log-transformed serum corticosterone, n = 2; water:NaCl consumption ratio across

weeks, n = 8) are provided in S33 Table. Graphs were created with GraphPad Prism, and show the mean ± the 95% confidence interval (CI), with non-repeated measures graphs also showing individual data points.

## Results

No shock controls were built into every Experiment to evaluate how consumption of a high salt (4.0% NaCl) diet, relative to the control (0.4% NaCl) diet, might affect fear behavior in the absence of contextual fear conditioning. For 'training' of no shock controls, no three-way interactions of time × sex × diet were observed in any of the three experiments, nor were two-way interactions of sex × diet nor time × diet (S1 Table). The only two-way interaction evaluated that reached the significance threshold was for a time × sex interaction in training for Experiment 3 (S1 Table), indicating that during 'training' one sex behaved differently over time than the other. In Experiments 1 and 2, no significant main effects of sex nor diet were detected, though both had significant main effects of time (S1 Table), meaning behavior of mice changed over the course of 'training'. This is to be expected when mice are initially exploring a new environment, then become complacent in the absence of any apparent threat. Pairwise comparisons indicated a few time points at which significant differences in freezing behavior between males consuming control and high salt diets were detected (S1A Fig, left and right panels), but these did not exhibit a consistent pattern, and were absent in Experiment 2 (S1A Fig, middle panel), suggesting they were not meaningful.

As with 'training', testing of no shock control mice revealed no significant interactions involving diet, and no main effect of diet (S2 Table), illustrating that diet did not impact freezing behavior in the absence of any fear conditioning. A significant interaction between Experiment × sex was observed, but Holm-Šídák's post-hoc tests revealed no significant differences across the sexes within any Experiment, nor within the sexes across any Experiments (S1B Fig), indicating at a granular level, there were no Experiment-specific sex differences.

Acquisition of contextual fear resulted in no significant three- nor two-way interactions in any of the three Experiments within either sex (S3–S5 Tables). Likewise, no main effects of Context nor diet were detected (S3–S5 Tables), meaning diet condition did not affect fear learning, and there were no inherent differences in mice to be tested in one Context or the other. In fact, the only significant main effect was that of time, detected across all three Experiments (S3–S5 Tables). This illustrates all mice that underwent contextual fear conditioning with foot shocks present during training exhibited increasing levels of freezing (i.e., fear behavior) over the course of the training session (Fig 2). Pairwise comparisons between male mice to be tested in the Neutral Context indicated that, in Experiment 2, mice consuming high salt exhibited elevated freezing levels following the second and third of the five training foot shocks, relative to mice consuming control diet (Fig 2B, right panel). All other acquisition curves lacked significant differences across diets and future testing contexts (Fig 2), indicating that a high salt diet does not impact acquisition of contextual fear after two (Experiments 1, 3) or six (Experiment 2) weeks of consumption.

Unlike context fear training, where diet was without significant effect on acquisition, high salt diet consumption did significantly affect context fear expression. Moreover, these dietary effects were often sex-selective. Across all three Experiments, there were no significant three-way interactions, nor were there any significant diet × context interactions (S7 Table). However, sex-specific fear behaviors between contexts were significant for Experiments 2 and 3, and fear behaviors across diet and sex combinations were significant for Experiment 3 (S7 Table). Further, diet alone had a significant effect in Experiment 2, and sex and context independently had significant effects on fear behavior in Experiment 1 (S7 Table). In other

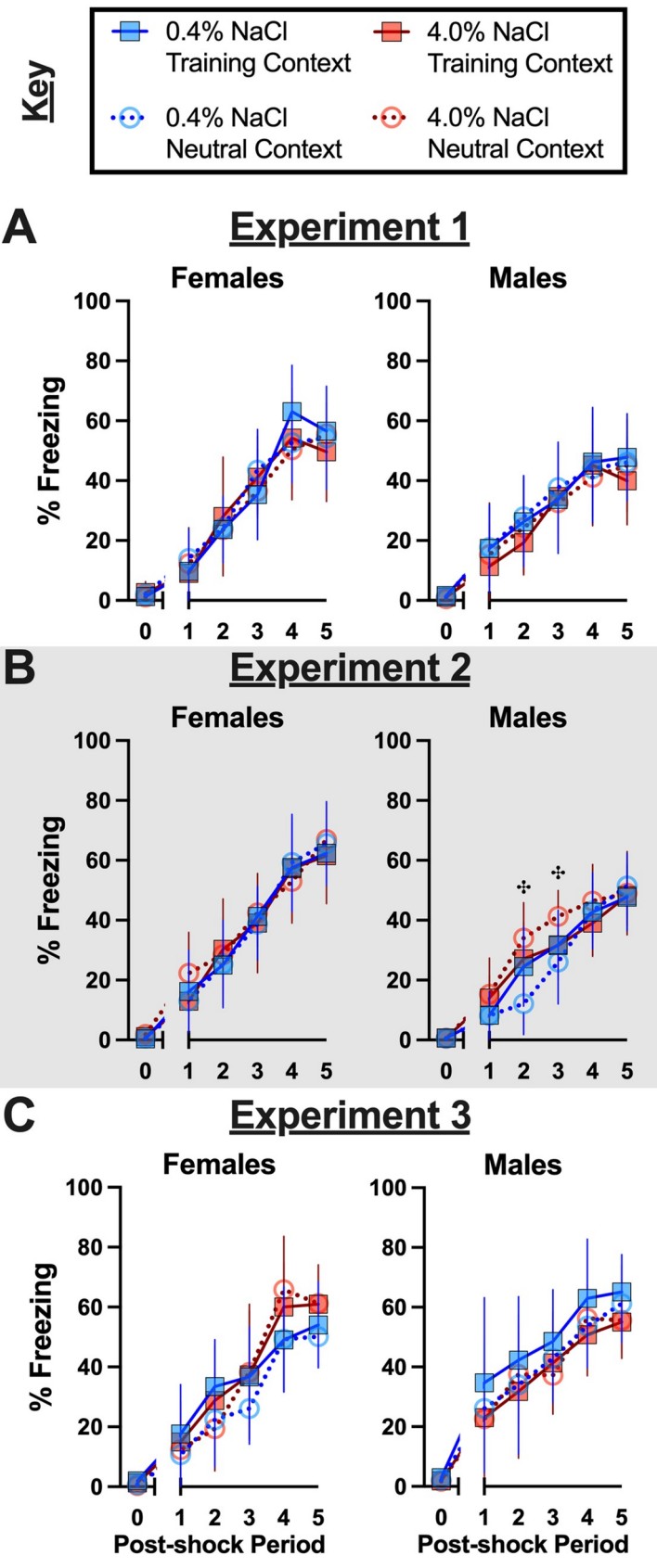

**Fig 2. Context fear training across Experiments.** Mice assigned to 0.4% NaCl represented by blue symbols, mice assigned to 4.0% NaCl represented by red symbols; mice to be tested in Training Context represented by squares and solid lines, mice to be tested in Neutral Context represented by circles and dotted lines. All mice were trained in the Training Context with five, pseudorandomized mild (1 s, 0.8 mA) foot shocks during a 6 min training session. Percent freezing during training of mice in A) Experiment 1, B) Experiment 2 (grey shading), and C) Experiment 3. Baseline percent freezing averaged across the first two minutes of the training session and plotted along x-axis as 0 time point. Average percent freezing for each 30 sec period following each of the five mild foot shocks are plotted along x-axis as Post-shock Periods 1–5. Experiment 1: 0.4% NaCl females Training Context, n = 7; 0.4% NaCl females Neutral Context, n = 9; 4.0% NaCl females Training Context, n = 8; 4.0% NaCl females Neutral Context, n = 9; 0.4% NaCl males Training Context, n = 8; 0.4% NaCl males Neutral Context, n = 9; 4.0% NaCl males Training Context, n = 7; 4.0% NaCl males Neutral Context, n = 8. Experiment 2: 0.4% NaCl females Training Context, n = 9; 0.4% NaCl females Neutral Context, n = 8; 4.0% NaCl females Training Context, n = 8; 4.0% NaCl females Neutral Context, n = 8; 0.4% NaCl males Training Context, n = 9; 0.4% NaCl males Neutral Context, n = 8; 4.0% NaCl males Training Context, n = 8; 4.0% NaCl males Neutral Context, n = 10. Experiment 3: 0.4% NaCl females Training Context, n = 8; 0.4% NaCl females Neutral Context, n = 9; 4.0% NaCl females Training Context, n = 9; 4.0% NaCl females Neutral Context, n = 8; 0.4% NaCl males Training Context, n = 7; 0.4% NaCl males Neutral Context, n = 8; 4.0% NaCl males Training Context, n = 8; 4.0% NaCl males Neutral Context, n = 8. Data are graphed as mean ± 95% confidence interval. **\*** indicates $p < 0.05$ difference between males to be tested in Neutral Context and consuming 0.4% and 4.0% NaCl diets at indicated time points.

words, diet was without a significant effect in Experiment 1, but diet did influence the outcomes of Experiments 2 and 3, and for all three Experiments, sex had a significant–and mostly interactional–impact as well.

Holm-Šídák's post-hoc testing helped indicate directionality of high salt consumption's effects on contextual fear expression and generalization across sexes for the three different Experiment timelines employed. Experiment 1 involved training mice after two weeks of diet manipulation, and testing 48 h after training. The expected low (average ~20% or less) freezing levels in the Neutral Context, with higher (average ~40% or more) freezing levels in the Training Context, were observed across nearly all the sex and diet combinations (Fig 3A). A nonsignificant trend was noted for contextual fear expression in the Training Context between females and males consuming the high salt diet (p = 0.0542; Fig 3A). Excepting this, Experiment 1 suggests that contextual fear expression after two weeks of high salt diet consumption is unaffected across contexts and sexes.

Experiment 2 involved training mice after six weeks of diet manipulation, then testing 48 h following training. Similar to Experiment 1, contextual fear expression was largely unaffected by diet or sex, with one exception. Male mice consuming the control diet exhibited reduced contextual fear expression in the Training Context as compared to female mice consuming the control diet (Fig 3B); this sex difference in Training Context fear expression was absent in mice of both sexes consuming the high salt diet. As anticipated when testing for contextual fear expression relatively soon after training (in this case, 48 h), freezing behavior averaging ≤ 20% indicated that mice did not exhibit fear behavior in the Neutral Context (see S1 Fig for no shock comparisons).

Experiment 3 evaluated how high salt intake affects contextual fear generalization by training mice after two weeks of diet manipulation, then testing for contextual fear expression four weeks later (i.e., after 6 wks of diet manipulation). When doing this, contextual fear generalization (freezing ~40% or more in Neutral Context) was observed in all but males consuming a high salt diet (Fig 3C). In fact, contextual fear expression in females consuming a high salt diet tested in the Neutral Context was so high that it was statistically indifferent from females consuming either diet and tested in the Training Context (Fig 3C). On the contrary, males tested in the Neutral Context after consuming high salt exhibited significantly lower contextual fear than females tested in the Neutral Context after consuming high salt (Fig 3C). This indicates a bidirectional, sex-selective effect of high salt consumption on contextual fear generalization.

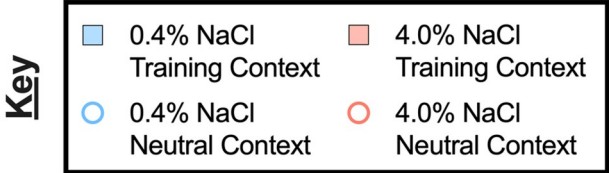

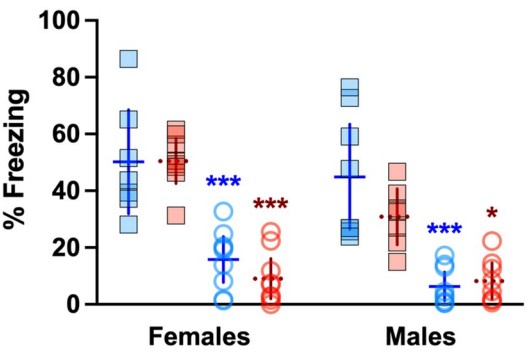

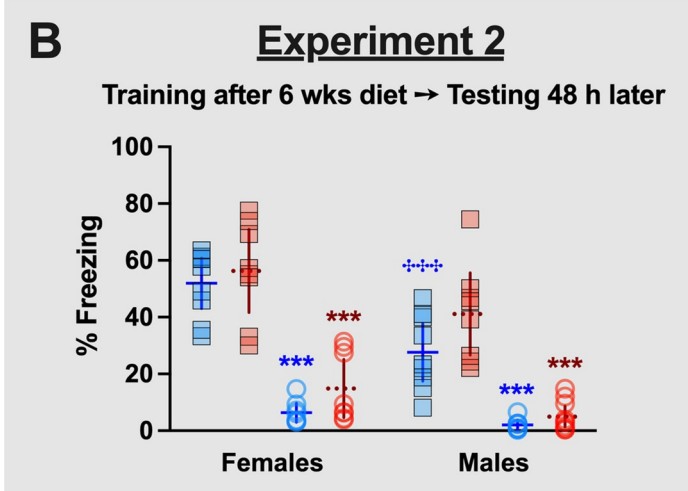

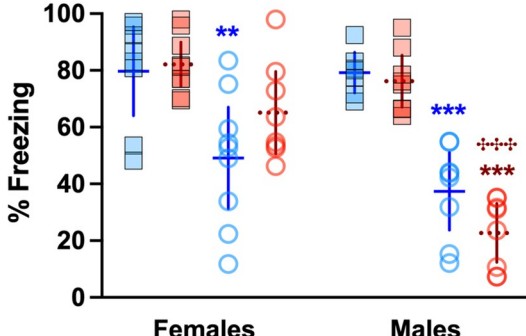

**Fig 3. Context fear expression across contexts and Experiments.** Mice assigned to 0.4% NaCl represented by blue symbols, mice assigned to 4.0% NaCl represented by red symbols; mice tested in Training Context represented by squares and solid lines, mice tested in Neutral Context represented by circles and dotted lines. Percent freezing during minutes two through six of the 10 min testing session are graphed for all mice. Testing occurred A) 48 h after training in Experiment 1, during which mice underwent two weeks of diet manipulation; B) 48 h after training in Experiment 2 (grey shading), during which mice underwent six weeks of diet manipulation; and C) four weeks after training in Experiment 3, during which mice underwent six total weeks of diet manipulation (training occurred after two weeks of diet manipulation). Experiment 1: 0.4% NaCl females Training Context, n = 7; 0.4% NaCl females Neutral Context, n = 9; 4.0% NaCl females Training Context, n = 8; 4.0% NaCl females Neutral Context, n = 9; 0.4% NaCl males Training Context, n = 8; 0.4% NaCl males Neutral Context, n = 9; 4.0% NaCl males Training Context, n = 7; 4.0% NaCl males Neutral Context, n = 8. Experiment 2: 0.4% NaCl females Training Context, n = 9; 0.4% NaCl females Neutral Context, n = 8; 4.0% NaCl females Training Context, n = 8; 4.0% NaCl females Neutral Context, n = 8; 0.4% NaCl males Training Context, n = 9; 0.4% NaCl males Neutral Context, n = 8; 4.0% NaCl males Training Context, n = 8; 4.0% NaCl males Neutral Context, n = 10. Experiment 3: 0.4% NaCl females Training Context, n = 8; 0.4% NaCl females Neutral Context, n = 9; 4.0% NaCl females Training Context, n = 9; 4.0% NaCl females Neutral Context, n = 8; 0.4% NaCl males Training Context, n = 7; 0.4% NaCl males Neutral Context, n = 8; 4.0% NaCl males Training Context, n = 8; 4.0% NaCl males Neutral Context, n = 8. Data are graphed as mean ± 95% confidence interval. *p<0.05, **p<0.01, ***p<0.001 indicate difference between Training Context and Testing Context within same sex and diet. ^***indicates p<0.001 difference between females and males on same diet and tested in same context.

Regardless of diet condition, all mice had *ad libitum* access to drinking water. Nonetheless, we evaluated if serum osmolality was affected across diet condition, sex, or testing Context in mice trained with foot shocks (Fig 4A–4C), as well as across diet, sex, and Experiment in control no shock mice. In control no shock mice, no three-way interactions were observed, but a significant two-way interaction where serum osmolality across the sexes differed between Experiments (S11 Table). Despite this interaction, Holm-Šídák's post-hoc testing revealed no significant directional differences when the data were examined between sexes within each Experiment for control no shock mice. Analyses of mice trained with foot shocks revealed zero three- or two-way interactions, and no significant main effects (S12 Table). Serum osmolality was not affected by diet, neither in control no shock mice nor in fear conditioned mice. Taken together, these findings suggest serum osmolality was unaltered in all mice, and thus the behavioral changes observed were not the consequence of osmotic stress. For food, water, salt, and kcal consumption data, plus body weight records, readers are referred to the Supporting Information.

In addition to serum osmolality, serum corticosterone levels were also measured in all mice 48 h after completion of context fear testing. Similar to serum osmolality, no three-way interactions were detected in log-transformed serum corticosterone (cort) levels of no shock control mice, but we did find a significant interaction between sexes across Experiments (S13 Table). Unlike with serum osmolality, Holm-Šídák's post-hoc tests revealed directional, diet-specific differences in cort levels within sexes across Experiments, and across sexes within an Experiment (S6 Fig). Specifically, control-diet consuming females had lower cort levels in Experiment 2 relative to Experiment 1 (p = 0.0423). Similarly, high salt-consuming males had lower cort levels in Experiment 3 relative to Experiment 1 (p = 0.0109). Within Experiment 3, females consuming high salt had significantly higher cort levels relative to high salt males (p = 0.0143). Considering these changes were observed in mice that were not exposed to a foot shock at any point, these data suggest that six weeks of high salt consumption could sex-selectively reduce cort levels in males, but not females. However, the absence of a significant difference (p = 0.2369) between cort levels of high salt males between Experiments 1 and 2 challenges this interpretation.

Cort levels in mice that underwent context fear conditioning with foot shocks had no significant three- nor two-way significant interactions (S14 Table). Sex alone had a significant effect on cort levels in Experiments 1 and 3, whereas context specifically influenced cort levels in Experiment 2. Drilling down to directional differences, there were largely no significant

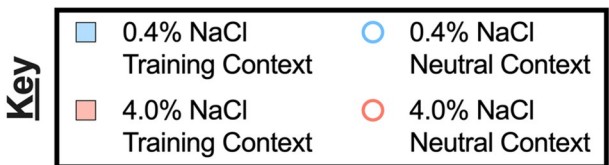

## A    Experiment 1

**Training after 2 wks diet → Testing 48 h later**

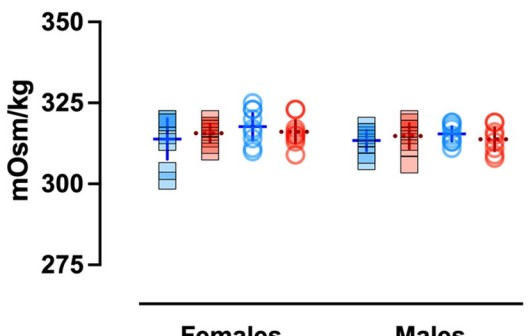

## B    Experiment 2

**Training after 6 wks diet → Testing 48 h later**

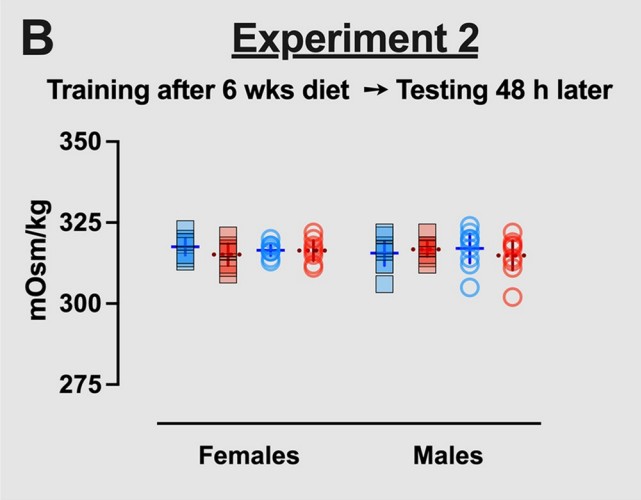

## C    Experiment 3

**Training after 2 wks diet → Testing 4 wks later**

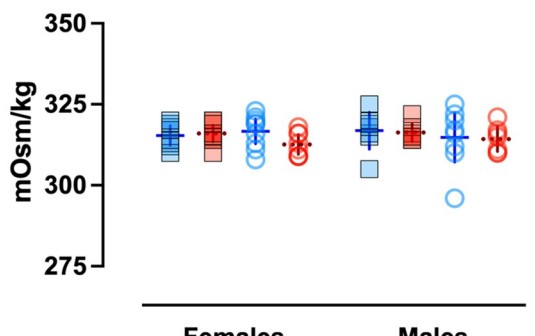

**Fig 4. Serum osmolality in context fear conditioned mice across Experiments.** Mice assigned to 0.4% NaCl represented by blue symbols, mice assigned to 4.0% NaCl represented by red symbols; mice tested in Training Context represented by squares and solid lines, mice tested in Neutral Context represented by circles and dotted lines. Serum osmolality in context fear conditioned mice from A) Experiment 1, B) Experiment 2 (grey shading), and C) Experiment 3. Experiment 1: 0.4% NaCl females Training Context, n = 8; 0.4% NaCl females Neutral Context, n = 9; 4.0% NaCl females Training Context, n = 8; 4.0% NaCl females Neutral Context, n = 9; 0.4% NaCl males Training Context, n = 8; 0.4% NaCl males Neutral Context, n = 9; 4.0% NaCl males Training Context, n = 8; 4.0% NaCl males Neutral Context, n = 8. Experiment 2: 0.4% NaCl females Training Context, n = 9; 0.4% NaCl females Neutral Context, n = 9; 4.0% NaCl females Training Context, n = 8; 4.0% NaCl females Neutral Context, n = 8; 0.4% NaCl males Training Context, n = 9; 0.4% NaCl males Neutral Context, n = 9; 4.0% NaCl males Training Context, n = 9; 4.0% NaCl males Neutral Context, n = 9. Experiment 3: 0.4% NaCl females Training Context, n = 8; 0.4% NaCl females Neutral Context, n = 9; 4.0% NaCl females Training Context, n = 9; 4.0% NaCl females Neutral Context, n = 8; 0.4% NaCl males Training Context, n = 7; 0.4% NaCl males Neutral Context, n = 8; 4.0% NaCl males Training Context, n = 7; 4.0% NaCl males Neutral Context, n = 7. Data are graphed as mean ± 95% confidence interval.

differences detected by Holm-Šídák's post-hoc tests (Fig 5A and 5B). The one exception was that in Experiment 3, males had lower cort levels than females (p = 0.0183), specifically between those consuming control diet and tested in the Training Context (Fig 5C). Similar, but non-significant, trends were noted between females and males tested in the Neutral Context consuming control diet (p = 0.0549) or consuming high salt (p = 0.0859; Fig 5C) for Experiment 3. Generally, cort levels did not consistently vary as a result of sex, diet, testing context, nor their interaction, indicating that shifts in circulating corticosterone are not responsible for the observed changes in contextual fear expression nor contextual fear generalization. Circulating cort levels were largely not correlated with contextual fear expression, except in females consuming high salt and tested in the Training Context in Experiment 3 (p = 0.039; see S23 Fig).

## Discussion

This is the first investigation into how context fear training, and subsequent expression and generalization, are affected across the sexes by a high salt diet. Our findings suggest that a relatively short (two week) period of high salt consumption might reduce expression of recently learned context fear in males, whereas a longer (six week) consumption period of high salt prevents decreases in expression of recently learned context fear in males. Moreover, context fear generalization was enhanced in females, but attenuated in males, by consumption of a high salt diet. Despite being contrary to our hypothesis regarding males, these data indicate that excess salt consumption affects behavior in a sex-selective manner. Further, these behavioral changes occurred independent of any changes in serum osmolality, and likewise largely did not map onto serum corticosterone levels, indicating that neither osmotic nor physiologic stress states were responsible.

Previous studies assessing how excess salt intake affects context fear behavior exclusively used male rats [25] or male mice [26]. These studies both used higher salt diets of 8.0% NaCl, compared to the 4.0% NaCl diet used here [25, 26]. Both teams reported attenuated context fear expression, indexed by freezing behavior. This freezing behavior was measured during testing that occurred in the Training Context either 24 h [26], or both 2 and 24 h [25], after training. Their observations differ from ours, which did not observe significant effects of high salt diet on contextual fear expression in the Training Context. A similarity between the present study and that of Ge and colleagues is the absence of any diet influence on fear acquisition to the auditory cue [26], or in our case, to the context. However, this raises an important caveat regarding both prior investigations—rodents were trained using a cued fear paradigm with an auditory tone [25, 26], rather than training the rodents as the present study did with a traditional context fear conditioning protocol that lacks discrete cues. These different training

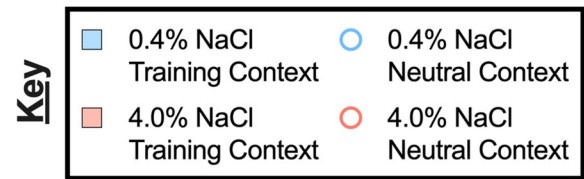

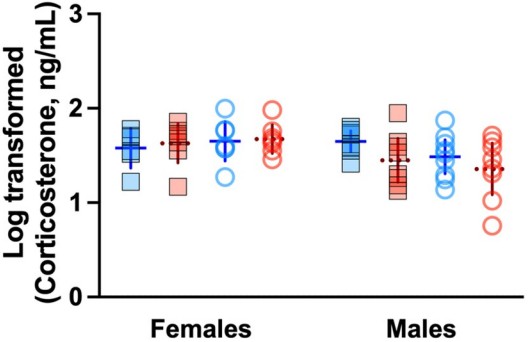

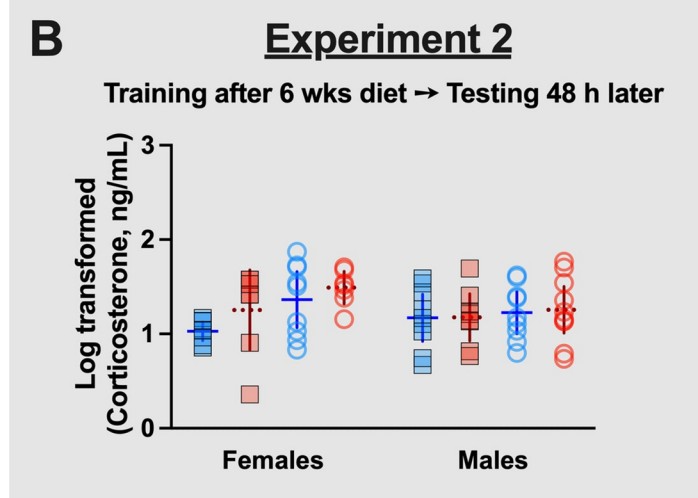

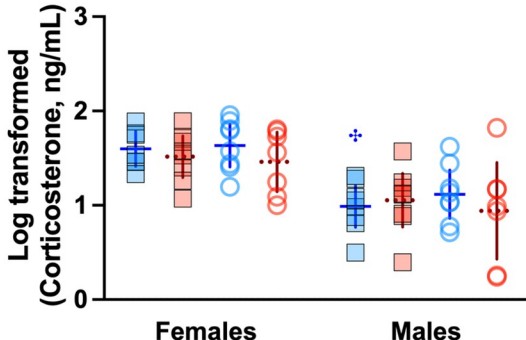

**Fig 5. Log-transformed serum corticosterone levels in fear conditioned mice across Experiments.** Mice assigned to 0.4% NaCl represented by blue symbols, mice assigned to 4.0% NaCl represented by red symbols; mice tested in Training Context represented by squares and solid lines, mice tested in Neutral Context represented by circles and dotted lines. Log-transformed serum corticosterone levels in context fear conditioned mice in A) Experiment 1, B) Experiment 2 (grey shading), and C) Experiment 3. Experiment 1: 0.4% NaCl females Training Context, n = 6; 0.4% NaCl females Neutral Context, n = 7; 4.0% NaCl females Training Context, n = 7; 4.0% NaCl females Neutral Context, n = 7; 0.4% NaCl males Training Context, n = 8; 0.4% NaCl males Neutral Context, n = 9; 4.0% NaCl males Training Context, n = 8; 4.0% NaCl males Neutral Context, n = 8. Experiment 2: 0.4% NaCl females Training Context, n = 8; 0.4% NaCl females Neutral Context, n = 9; 4.0% NaCl females Training Context, n = 7; 4.0% NaCl females Neutral Context, n = 7; 0.4% NaCl males Training Context, n = 9; 0.4% NaCl males Neutral Context, n = 9; 4.0% NaCl males Training Context, n = 8; 4.0% NaCl males Neutral Context, n = 10. Experiment 3: 0.4% NaCl females Training Context, n = 7; 0.4% NaCl females Neutral Context, n = 8; 4.0% NaCl females Training Context, n = 8; 4.0% NaCl females Neutral Context, n = 7; 0.4% NaCl males Training Context, n = 8; 0.4% NaCl males Neutral Context, n = 8; 4.0% NaCl males Training Context, n = 8; 4.0% NaCl males Neutral Context, n = 7. Data are graphed as mean ± 95% confidence interval. *indicates p<0.05 difference between females and males on same diet and tested in same context.

procedures engage distinct brain regions [49, 50], and likewise elicit distinguishable patterns of neuronal activation [51–53]. Thus, in the present study we assessed contextual fear expression specifically following contextual fear conditioning, rather than cued fear conditioning.

Generalization of contextual fear is a hallmark symptom of many anxiety disorders, and is a conserved cross-species neurobehavioral phenomenon that is useful for understanding how neurophysiological changes influence behaviors [32, 54]. The majority of contextual fear generalization investigations in rodents have been performed in males [33], though studies have recently begun to include females. Such sex-inclusive studies indicate that rodents of both sexes can and do exhibit generalized contextual fear, at timepoints both days [55] and weeks [56] after context fear training. Sex differences in the influence of various contextual modalities have been reported (see review [33]), and in our study contextual differences included tactile, olfactory, and visual modalities. We observed robust discrimination between contexts across the sexes when training was temporally proximal to testing, as well as convincing evidence of contextual fear generalization in both sexes consuming control diet when testing occurred four weeks following training. In contrast to our hypothesis that high salt intake would enhance contextual fear generalization in males, we instead found an attenuation of this measure in males, but an enhancement in females. Given the near absence of investigations into how excess salt intake affects female behaviors (see review [28]), plus the relatively scant literature on contextual fear generalization in females (see review [33]), we did not have enough information to formulate an *a priori* hypothesis for females consuming high salt. Some researchers have suggested excess salt intake exacerbates neuropsychiatric symptoms and conditions [29, 30]. Our data support this relationship, at least in females. However, the relationship between high salt consumption and anxiety-relevant behaviors was not supported in our male mice, and in fact our data suggest an inverse relationship. Given frequent comorbidity between cardiovascular diseases and pathological anxiety [11–15] (see also reviews [16, 17]), and the adverse impact of overconsumption of salt on cardiovascular disease risk [5, 10], this finding in males was surprising. Nonetheless, it may be that in males, high salt-mediated cardiovascular pathology is necessary for emergence of increased anxiety-related behaviors, whereas in females excess salt is sufficient. This would parallel findings indicating greater metabolic disruptions in men relative to women with comorbid obesity and mental health disorders [57–59] (see review [60]), greater incidence of depression and anxiety in men versus women with multiple sclerosis [61], and increased incidence of a biomarker of cardiovascular disease in men versus women with depression [62]. However, some literature suggests this is not a consistent finding across disease conditions [63].

Along these lines, we evaluated if mice consuming excess salt exhibited peripheral indicators of osmotic (serum osmolality) or physiological (serum corticosterone) stress. Changes in

these measures could suggest an overall disrupted state of homeostasis that might confound contextual fear learning and/or expression. For example, osmotic stress can reduce sensitivity to pain [64]. Though the shock level (0.8 mA) and duration (1 sec) that we use are intended to only elicit discomfort/distress, it remains possible that osmotic stress could skew sensory perception of this brief aversive experience. Similarly, endogenous corticosterone levels can influence temporal development of contextual fear generalization as well as the persistence of contextual discrimination [65]. The absence of any sex-, diet-, or context-specific differences in serum osmolality indicates that osmotic stress was not present in our experimental animals. No significant correlations between cort levels and fear expression in the Neutral Context were detected for either sex in Experiment 3. Indeed, of all the mice trained in the Training Context with foot shocks, only Experiment 3 females consuming high salt and tested in the Training Context displayed a significant (negative) correlation between context fear expression and cort levels (S23 Fig). The absence of any significant correlations between cort levels and any other sex/diet/context combination across Experiments indicates that cort levels overall probably did not confound observed context fear expression and generalization.

## Limitations

Limitations of this study include singly housing of mice to allow for tracking of individual food/water/salt/kcal consumption and changes in body weight, meaning that mice were excluded from social stimulation. Some data suggests behavioral responses to single housing differs across the sexes in mice and rats (see review [66]), so this might have influenced sex-specific behaviors observed in the present study. However, a meta-analysis of 293 male and female mouse studies indicates that singly housing mice reduces coefficients of variation in trait variability by 37% in both sexes [67], suggesting our housing condition may have reduced the variability of our data. Diet manipulations were relatively short (two or six weeks), restricted to non-breeding adults, and contained no other accompanying unhealthy dietary components (e.g., low fiber, high fat). Only a single shock level and context fear conditioning paradigm was used, and mice were only tested in a single context. Thus, for future studies it will be important to evaluate behavior in group housed mice, assess longer diet manipulation periods and include additional unhealthy dietary components, and study how consumption of excess salt by younger mice affects their fear behaviors. Testing of behavior during the natural activity period of mice would be informative, as would assessing generalization and concurrent cort levels during different stages of the circadian cycle [68].

## Conclusions

Our study provides foundational evidence of sex-selective consequences of excess salt intake on context fear expression and generalization. Additional studies will be essential to flesh out these initial findings and determine what brain cell populations are most impacted by excess salt intake, and how the effects of high salt might be mitigated. One logical starting point is evaluation of microglia, given converging evidence that both fear processing [69, 70] and high salt intake [42, 71, 72] affect microglial activation. Moreover, oral administration of the brain-penetrant anti-inflammatory drug minocycline can mitigate microglial activation and behavioral changes in rodents consuming high salt [42], and the same drug attenuated fear measures in human participants [73]. Future studies should also assess how excess salt intake affects cued fear expression and discrimination, and in so doing begin identifying how specific limbic brain regions might differentially be affected in their functionality and/or connectivity following extended periods of excess salt consumption. The evidence presented here, in conjunction with past literature focused on salt's effects on behaviors outside of the fear domain(reviewed

in [28]), indicates that salt is an underappreciated, non-caloric influence on conserved mammalian behaviors.

## Supporting information

**S1 Table. Three-way repeated measures ANOVAs on context fear 'training' for control no shock mice across Experiments.**
(PDF)

**S2 Table. Three-way ANOVAs on context fear testing of control no shock mice across Experiments.**
(PDF)

**S3 Table. Three-way repeated measures ANOVAs on context fear training for context fear conditioned mice of both sexes in Experiment 1.**
(PDF)

**S4 Table. Three-way repeated measures ANOVAs on context fear training for context fear conditioned mice of both sexes in Experiment 2.**
(PDF)

**S5 Table. Three-way repeated measures ANOVAs on context fear training for context fear conditioned mice of both sexes in Experiment 3.**
(PDF)

**S6 Table. Three-way repeated measures ANOVAs on full 10 min time course of context fear testing for control no shock mice across Experiments.**
(PDF)

**S7 Table. Three-way ANOVAs on context fear expression (min 2–6) for context fear conditioned mice across Experiment.**
(PDF)

**S8 Table. Three-way repeated measures ANOVAs on full 10 min time course of context fear testing for mice of both sexes in Experiment 1.**
(PDF)

**S9 Table. Three-way repeated measures ANOVAs on full 10 min time course of context fear testing for mice of both sexes in Experiment 2.**
(PDF)

**S10 Table. Three-way repeated measures ANOVAs on full 10 min time course of context fear testing for mice of both sexes in Experiment 3.**
(PDF)

**S11 Table. Three-way ANOVAs on serum osmolality in control no shock mice across Experiments.**
(PDF)

**S12 Table. Three-way ANOVAs on serum osmolality in context fear conditioned mice across Experiments.**
(PDF)

**S13 Table. Three-way ANOVAs on log-transformed serum corticosterone levels in control no shock mice across Experiments.**
(PDF)

**S14 Table. Three-way ANOVAs on log-transformed serum corticosterone in context fear conditioned mice across Experiments.**
(PDF)

**S15 Table. Three-way repeated measures ANOVAs on weekly average water consumption per day for context fear conditioned mice across Experiments.**
(PDF)

**S16 Table. Three-way repeated measures ANOVAs on weekly average food consumption per day for context fear conditioned mice across Experiments.**
(PDF)

**S17 Table. Three-way repeated measures ANOVAs on weekly body weight change for context fear conditioned mice across Experiments.**
(PDF)

**S18 Table. Three-way repeated measures ANOVAs on twice weekly body weight measurements of context fear conditioned mice across Experiments.**
(PDF)

**S19 Table. Three-way repeated measures ANOVAs on weekly average water consumption per day for control no shock mice across Experiments.**
(PDF)

**S20 Table. Three-way repeated measures ANOVAs on weekly average food consumption per day for control no shock mice across Experiments.**
(PDF)

**S21 Table. Three-way repeated measures ANOVAs on weekly body weight changes for control no shock mice across Experiments.**
(PDF)

**S22 Table. Three-way repeated measures ANOVAs on twice weekly body weights of control no shock mice across Experiments.**
(PDF)

**S23 Table. Three-way repeated measures ANOVAs on weekly average NaCl consumed per day by context fear conditioned mice across Experiments.**
(PDF)

**S24 Table. Three-way repeated measures ANOVAs on weekly average NaCl consumed as a percentage of body weight in context fear conditioned mice across Experiments.**
(PDF)

**S25 Table. Three-way repeated measures ANOVAs on weekly ratio of water to NaCl consumed in context fear conditioned mice across Experiments.**
(PDF)

**S26 Table. Three-way repeated measures ANOVAs on weekly average kcal consumed per day by context fear conditioned mice across Experiments.**
(PDF)

**S27 Table. Three-way repeated measures ANOVAs on weekly kcal consumed as a percentage of body weight in context fear conditioned mice across Experiments.**
(PDF)

**S28 Table. Three-way repeated measures ANOVAs on weekly average NaCl consumed per day by control no shock mice across Experiments.**
(PDF)

**S29 Table. Three-way repeated measures ANOVAs on weekly NaCl consumed as a percentage of body weight by control no shock mice across Experiments.**
(PDF)

**S30 Table. Three-way repeated measures ANOVAs on weekly water to NaCl ratio consumed by control no shock mice across Experiments.**
(PDF)

**S31 Table. Three-way repeated measures ANOVAs on weekly average kcal consumed per day by control no shock mice across Experiments.**
(PDF)

**S32 Table. Three-way repeated measures ANOVAs on weekly average kcal consumed as a percentage of body weight by control no shock mice across Experiments.**
(PDF)

**S33 Table. Details of confirmed outliers ($>$ 5 standard deviations (SD) ± average of data set sans suspected outlier) within respective data sets.** Outliers are marked with red font.
(PDF)

**S1 Fig. Fear behavior in no shock control mice across Experiments.**
(PDF)

**S2 Fig. Fear behavior in no shock control mice across Experiments with shortened Y axes.**
(PDF)

**S3 Fig. Time course of context fear expression testing in no shock groups.**
(PDF)

**S4 Fig. Time course of context fear expression testing in shock groups.**
(PDF)

**S5 Fig. Serum osmolality in no shock control mice across Experiments.**
(PDF)

**S6 Fig. Log-transformed serum corticosterone levels of no shock control mice across Experiments.**
(PDF)

**S7 Fig. Raw (pre-transformed) serum corticosterone levels of no shock control mice across Experiments.**
(PDF)

**S8 Fig. Raw (pre-transformed) serum corticosterone levels in fear conditioned mice across Experiments.**
(PDF)

**S9 Fig. Average water consumed per day by context fear conditioned mice across Experiments.**
(PDF)

**S10 Fig. Average food consumed per day by context fear conditioned mice across Experiments.**
(PDF)

**S11 Fig. Weekly body weight change in context fear conditioned mice across Experiments.**
(PDF)

**S12 Fig. Body weight of context fear conditioned mice across Experiments.**
(PDF)

**S13 Fig. Average water and food consumed per day by control no shock mice across Experiments.**
(PDF)

**S14 Fig. Body weights and body weight changes in control no shock mice across Experiments.**
(PDF)

**S15 Fig. Average NaCl consumed per day by context fear conditioned mice across Experiments.**
(PDF)

**S16 Fig. NaCl consumed as a percentage of body weight by context fear conditioned mice across Experiments.**
(PDF)

**S17 Fig. Water to NaCl consumed ratio for context fear conditioned mice across Experiments.**
(PDF)

**S18 Fig. Average kcal consumed per day by context fear conditioned mice across Experiments.**
(PDF)

**S19 Fig. Consumed kcal as a percentage of body weight by context fear conditioned mice across Experiments.**
(PDF)

**S20 Fig. Average NaCl consumed per day and NaCl consumed as a percentage of body weight by control no shock mice across Experiments.**
(PDF)

**S21 Fig. Ratio of water to NaCl consumed by control no shock mice across Experiments.**
(PDF)

**S22 Fig. Average kcal consumed per day and kcal consumed as a percentage of body weight by control no shock mice across Experiments.**
(PDF)

**S23 Fig. Correlations between context fear expression and log-transformed serum corticosterone levels across contexts and Experiments.**
(PDF)

**S24 Fig. Correlations between context fear expression and log-transformed serum corticosterone levels in control no shock mice across Experiments.**
(PDF)

## Acknowledgments

We gratefully acknowledge the mice used in this study, and the expert care thereof by our caretakers. Figures were created with Biorender.com (Toronto ON) and GraphPad Prism (San Diego, CA).

## Author Contributions

**Conceptualization:** T. Lee Gilman.

**Data curation:** Jasmin N. Beaver, Brady L. Weber, Matthew T. Ford, T. Lee Gilman.

**Formal analysis:** T. Lee Gilman.

**Funding acquisition:** Jasmin N. Beaver, Sarah K. Kassis, T. Lee Gilman.

**Investigation:** Jasmin N. Beaver, Brady L. Weber, Matthew T. Ford, Anna E. Anello, Kaden M. Ruffin, Sarah K. Kassis, T. Lee Gilman.

**Methodology:** Jasmin N. Beaver, Brady L. Weber, Matthew T. Ford, Anna E. Anello, Kaden M. Ruffin, Sarah K. Kassis, T. Lee Gilman.

**Project administration:** Jasmin N. Beaver, Brady L. Weber, T. Lee Gilman.

**Resources:** T. Lee Gilman.

**Software:** T. Lee Gilman.

**Supervision:** T. Lee Gilman.

**Validation:** Jasmin N. Beaver, Brady L. Weber, T. Lee Gilman.

**Visualization:** Jasmin N. Beaver, T. Lee Gilman.

**Writing – original draft:** T. Lee Gilman.

**Writing – review & editing:** Jasmin N. Beaver, Brady L. Weber, Matthew T. Ford, Anna E. Anello, Kaden M. Ruffin, Sarah K. Kassis, T. Lee Gilman.

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
