## [Decision Letter · Decision Letter 0]

7 Mar 2023

PONE-D-22-32310Generalization of contextual fear is sex-specifically affected by high salt intakePLOS ONE

Dear Dr. Gilman,

Thank you for submitting your manuscript to PLOS ONE. After careful consideration, we feel that it has merit but does not fully meet PLOS ONE’s publication criteria as it currently stands. Therefore, we invite you to submit a revised version of the manuscript that addresses the points raised during the review process.

We look forward to receiving your revised manuscript.

Kind regards,

Thiago Fernandes, PhD

Academic Editor

PLOS ONE

Journal Requirements

Funding for this work was provided by Kent State University, and the Applied Psychology Center in the Department of Psychological Sciences at Kent State University.  

Funding for this work was provided by Kent State University, and the Applied Psychology Center in the Department of Psychological Sciences at Kent State University.  

However, funding information should not appear in the Acknowledgments section or other areas of your manuscript. We will only publish funding information present in the Funding Statement section of the online submission form. 

Funding for this work was provided by Kent State University, and the Applied Psychology Center in the Department of Psychological Sciences at Kent State University.  

Additional Editor Comments:

Please respond to each point AND highlight all changes.

Reviewers' comments:

Reviewer's Responses to Questions

**Comments to the Author**

1. Is the manuscript technically sound, and do the data support the conclusions?

Reviewer #1: Yes

Reviewer #2: Partly

2. Has the statistical analysis been performed appropriately and rigorously? 

Reviewer #1: Yes

Reviewer #2: Yes

3. Have the authors made all data underlying the findings in their manuscript fully available?

Reviewer #1: Yes

Reviewer #2: Yes

4. Is the manuscript presented in an intelligible fashion and written in standard English?

Reviewer #1: Yes

Reviewer #2: Yes

5. Review Comments to the Author

Reviewer #1: This is an interesting study. The following issues should be addressed.

1. Fig. 2. Adjust the Y axis to reflect the low freezing levels seen. Also, it would be better to show these data as bar graphs so that group differences can be appreciated.

2. Only 5 seconds post the shock are analyzed. It seems better to analyze the largest ISI possible.

3. Females and males respond different to singly housing. Therefore, it is conceivable that in experiment 3 sex-dependent effects of singly housing might have contributed to the sex-dependent effects seen. This should be acknowledged.

4. It seems better to move figure panels not showing any significance from the main figure to supplementary figures. Some figure, like figure 6, could be a main or supplementary table as well. Fig. 4 is the main figure and receives less attention now.

5. There seems more of a sex difference in the training context in Experiments 1 and 2 in Fig. 4 than in Experiment 3. One wonder whether this might have contributed to what is seen in Experiment 3. In females, the pattern in the neutral context seems definitely similar in Experiments 2 and 3.

6. The salt effect in females in the neutral context seems subtle and not different between the two salt diet groups. The salt effect in males in the neutral context seems more pronounced.

7. It would be better to show the real cort data so that others can compare with other designs.

8. It would be better to remove p values from the figures, especially if they do not reach significance.

9. The text in the results section is written very stats heavy. While all correct, this makes it harder to read for those less versed in statistical interactions. The authors are encouraged to rephrase some of the text to make this manuscript more readable. The tables with the stats data are good and allow focusing less on stats result in the text.

Reviewer #2: This manuscript evaluates that possibility that a forced, high-salt diet affects contextual fear acquisition, recall and generalization in male and female C57BL/6J mice. I think there is some important data here, and it should be seen. Most of my concerns are with how the data, statistics, etc. are presented. Given that this is a straight-forward experiment, it would benefit to attempt to streamline what is being presented to the reader. I found myself paying too much attention to small details the were being presented/suggested rather than being able to focus on the main message (or two) as I read through the Results and Discussion presentation.

Specific comments:

Introduction: Lines 68-70 – I think it would be useful to provide some specific detail on how hippocampus/amygdala are affected by high-salt.

Consider providing a clear “N per group/sex” in the Methods.

Describe in more detail why 5x shock was used. 1 or two is sufficient to engender contextual fear.

Page 8, Line 158 – Much more thorough justification needs to be provided for why the authors are selectively presenting/analyzing data from minutes 2-6 out of the 10-minute test. Without this, it appears that the data has been “cherry picked” to show some effects.

Figure 2 – It is confusing to refer to a “post-shock period” when these experiments did not employ shock. Also, I find using the sex-symbols as data points to be distracting. Could you more simply use blue/red squares and circles?

Consider placing the tables in a data supplement. There are 10 of them, several with multiple parts, and they are essentially all ANOVA tables.

Page 26, Line 526 – The overall statistical analyses appear to be rigorous. The notion of a “trend” presented in the Discussion detracts from this.

Page 27, Line 548 – The above is also true for the notion of a “bimodal” distribution, which would be very difficult to show with that number of data points.

Page 28, Line 579 – Same for “non-significant trends”

How do the authors know that 0.8mA is not painful?

While I think the authors are likely correct about the lack of social simulation and light-phase testing in their experiment, bringing this up distracts from the overall message regarding the data at hand. For example, it begs the question why reverse-phase lighting was not used in the first place?

Page 28, Line 566 – Sentence beginning “Nonetheless…”. I would like the authors to expand more upon this. From a biological standpoint, I’m left hanging and wondering “how would this be possible?”. Perhaps the authors can draw on other work for examples outside of hypertension/anxiety.

Do the authors think their findings would generalize to more classic mouse tests of “anxiety”, such as EPM or open field?

6. PLOS authors have the option to publish the peer review history of their article (what does this mean?). If published, this will include your full peer review and any attached files.

Reviewer #1: No

Reviewer #2: No

---

## [Author Response · Author response to Decision Letter 0]

20 Apr 2023

Please see uploaded file detailing all responses to Reviewers.

---

## [Decision Letter · Decision Letter 1]

11 May 2023

Generalization of contextual fear is sex-specifically affected by high salt intake

PONE-D-22-32310R1

Dear Dr. Gilman,

We’re pleased to inform you that your manuscript has been judged scientifically suitable for publication and will be formally accepted for publication once it meets all outstanding technical requirements.

Kind regards,

Thiago Fernandes, PhD

Academic Editor

PLOS ONE

Reviewers' comments:

Reviewer's Responses to Questions

**Comments to the Author**

1. If the authors have adequately addressed your comments raised in a previous round of review and you feel that this manuscript is now acceptable for publication, you may indicate that here to bypass the “Comments to the Author” section, enter your conflict of interest statement in the “Confidential to Editor” section, and submit your "Accept" recommendation.

Reviewer #1: All comments have been addressed

Reviewer #2: All comments have been addressed

2. Is the manuscript technically sound, and do the data support the conclusions?

Reviewer #1: Yes

Reviewer #2: Yes

3. Has the statistical analysis been performed appropriately and rigorously? 

Reviewer #1: Yes

Reviewer #2: Yes

4. Have the authors made all data underlying the findings in their manuscript fully available?

Reviewer #1: Yes

Reviewer #2: Yes

5. Is the manuscript presented in an intelligible fashion and written in standard English?

Reviewer #1: Yes

Reviewer #2: Yes

6. Review Comments to the Author

Reviewer #1: The authors did a fine job addressing the raised concerns. No additional edits are requested from this reviewer.

Reviewer #2: (No Response)

7. PLOS authors have the option to publish the peer review history of their article (what does this mean?). If published, this will include your full peer review and any attached files.

Reviewer #1: No

Reviewer #2: No

---

## [Editor Report · Acceptance letter]

16 May 2023

PONE-D-22-32310R1 

Generalization of contextual fear is sex-specifically affected by high salt intake 

Dear Dr. Gilman:

I'm pleased to inform you that your manuscript has been deemed suitable for publication in PLOS ONE. Congratulations! Your manuscript is now with our production department. 

Kind regards, 

on behalf of

Dr. Thiago P. Fernandes 

Academic Editor

PLOS ONE